# SINGAPO: SINGLE IMAGE CONTROLLED GENERATION OF ARTICUALTED PARTS IN OBJECTS

**Jiayi Liu[1], Denys Iliash[1], Angel X. Chang[1,2], Manolis Savva[1], Ali Mahdavi-Amiri[1]**
[1]Simon Fraser University, [2]Canada-CIFAR AI Chair, Amii
`3dlg-hcvc.github.io/singapo`

## ABSTRACT

We address the challenge of creating 3D assets for household articulated objects from a single image. Prior work on articulated object creation either requires multi-view multi-state input, or only allows coarse control over the generation process. These limitations hinder the scalability and practicality for articulated object modeling. In this work, we propose a method to generate articulated objects from a single image. Observing the object in resting state from an arbitrary view, our method generates an articulated object that is visually consistent with the input image. To capture the ambiguity in part shape and motion posed by a single view of the object, we design a diffusion model that learns the plausible variations of objects in terms of geometry and kinematics. To tackle the complexity of generating structured data with attributes in multiple domains, we design a pipeline that produces articulated objects from high-level structure to geometric details in a coarse-to-fine manner, where we use a part connectivity graph and part abstraction as proxies. Our experiments show that our method outperforms the state-of-the-art in articulated object creation by a large margin in terms of the generated object realism, resemblance to the input image, and reconstruction quality.

## 1 INTRODUCTION

Articulated objects are prevalent in our daily environments. Creating 3D assets that represent articulated objects is becoming increasingly important for building realistic and interactive virtual environments, which is essential for various indoor applications in robotics and embodied AI (Shen et al., 2021; Li et al., 2021; Puig et al., 2023; Li et al., 2024a; Kim et al., 2024). The creation of articulated assets typically requires manual labor from experts, which is time-consuming and expensive. Thus, it is desirable to automate this process to scalably enrich virtual environments.

A line of recent works reconstructs articulated objects from images or videos that capture the object from multi-views or multiple articulated states (Jiang et al., 2022; Liu et al., 2023a; Heppert et al., 2023; Song et al., 2024; Weng et al., 2024; Mandi et al., 2024). These works require careful alignment across views or across states, or recording the object in motion, which is not always feasible in practice. The challenges in acquisition of input data limit the scalability of these methods to real-world applications. Another line of work aims to generate articulated objects either unconditionally or guided by high-level constraints (Lei et al., 2023; Liu et al., 2024c). One of the primary challenges in content creation is providing users with the flexibility to specify the objects precisely. However, although object images are widely available and serve as intuitive guidance, their usage to guide generating articulated objects is under-explored. In this work, we propose a method for generating articulated household objects from a single image to bridge this gap.

Creating articulated assets from an image is challenging when: 1) the object is observed in the closed state where occlusions introduce ambiguity in the part shape and articulation; 2) the part structures are complex and diverse within and between object categories such as cabinets, desks, and refrigerators; 3) the parts are small and thin, making them hard to perceive from a single image. In this paper, we aim to tackle the task in the challenging setting where the input is an RGB image of an object in the resting state observed from a random view, and the output is the 3D asset of the articulated object. Our output consists of: 1) a graph specifying the part connectivity and kinematic hierarchy; 2) articulation parameters specifying the type, axis, and range of the joints connecting the parts; and 3) geometry of each articulated part to the level of actionable part (e.g., handles, knobs).

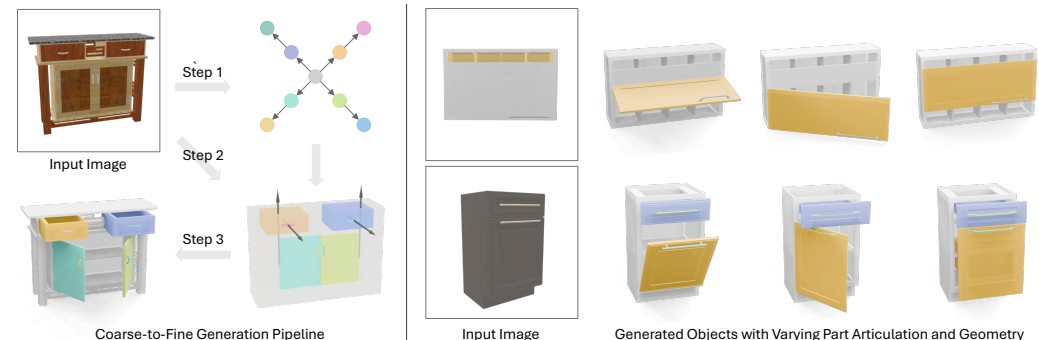

Figure 1: Our work proposes to generate 3D articulated objects from a single image observing the object in the resting state from a random view. **Left**: We design a pipeline to synthesize articulated assets progressively from coarse to fine details in a modular way. **Right**: Our method generates objects with varying part geometry and motion to account for the ambiguity in the input image.

To address these challenges, we propose to create 3D assets of articulated household objects from a single image using a generative model paradigm. We believe this strategy offers two key advantages. First, when an input image can be interpreted in various ways, a generative model produces multiple plausible solutions while ensuring consistency with the visual input, effectively capturing any ambiguity in the image as shown in Figure 1. Second, learning the distribution of articulated objects, a generative model bridges the gap between 2D and 3D by leveraging prior knowledge from the learned data distribution. This enables the model to more effectively handle input images of objects from arbitrary viewpoints to assemble a coherent 3D object, in contrast to regression-based methods from prior work that treat each part region independently (Chen et al., 2024b).

Addressing this problem necessitates a multi-level understanding of the object image, including recognizing all the articulated parts, inferring how these parts are connected, accurately recovering their 3D arrangement, imagining how each part articulates, and reconstructing the geometry of each part. To tackle this complexity, we propose a pipeline that decomposes the task into three progressive steps, synthesizing the articulated assets from coarse to fine details (see Figure 1 left). These steps include: 1) leveraging large vision-language models to infer the part connectivity graph from the image; 2) generating attributes to describe the articulated parts at the abstract level, guided by the graph and the image; and 3) collecting meshes from a part shape library to assemble the final object based on the outputs from previous steps. This modular design makes our method more efficient and enables easier user interventions to edit and refine the results. Additionally, this breakdown ensures that core understanding occurs at the appropriate level of abstraction in the second step. Specifically, we propose a transformer-based diffusion model that learns the spatial layout of the parts through image cross-attention, captures the interplay between part motion and shape via self-attention, and structures the parts using a graph with masked attention.

In summary, our main contributions are as follows: 1) to the best of our knowledge, we are the first to explore the task of generating articulated objects from a single image; 2) we design a pipeline to create articulated objects from coarse to fine details in a modular way, which is more interpretable and editable for users; 3) we propose a diffusion-based model to generate objects that are kinematically plausible and visually consistent with the input image, while allowing for variations to account for any ambiguity in the image; 4) we demonstrate our better reconstruction quality and generalization compared to the prior methods through systematic evaluations on multiple datasets.

## 2 RELATED WORK

**Generation of structured data.** Our task is closely related to the generation of structured data (Chaudhuri et al., 2020). The generation of 3D shapes with semantic parts is a widely studied problem with the main goal of modeling objects with geometric details and semantic grouping at the part level. Prior work either synthesizes objects in voxels with semantic labels (Wang et al., 2018; Li et al., 2020; Wu et al., 2020) or further considers the spatial structure, such as symmetry and support relationship, by jointly learning in the latent space (Wu et al., 2019) or explicitly modeling

a tree structure (Li et al., 2017; Mo et al., 2019; Gao et al., 2019). 3D assembly is a similar problem that aims to place primitives (Gadelha et al., 2020; Paschalidou et al., 2021; Jones et al., 2020; Xu et al., 2024), joints (Willis et al., 2022; Li et al., 2024b), or parts (Zhan et al., 2020; Li et al., 2020; Narayan et al., 2022; Xu et al., 2023; Koo et al., 2023) to form a shape. Scene generation work learns to compose scenes from objects by considering the spatial arrangement and relationship between objects (Wang et al., 2021; Wei et al., 2023; Tang et al., 2024). Similarly, building structure generation focuses on the general layout of the room or house (Nauata et al., 2020; 2021; Shabani et al., 2023; Tang et al., 2023). Articulated objects are a type of structured data that considers both part geometry and articulation in 3D. Thus articulated object generation poses a unique challenge (Liu et al., 2024b). In this work, we focus on image-conditioned generation of articulated 3D objects.

**Conditional generation on 3D structures.** Enabling control over the generation process is crucial for practical applications. This becomes more challenging for structured data generation, due to the need for fine-grained control and parsing the conditioning into structured components. Part-based object generation has been studied in settings driven by a partial shape to complete the object part-by-part or guided by an image (Wu et al., 2020; Niu et al., 2018). These methods only work with specific categories of objects that share similar structures. The recent advances in large-language models and generative models promote conditional generation from 2D (Yang et al., 2021; Rombach et al., 2022; Zhang et al., 2023a; Blattmann et al., 2023) to 3D with structures (Liu et al., 2024a). Many works on 3D scene generation have explored the control of object arrangements via text for room and object specification (Zhang et al., 2023b; Lin & Mu, 2024; Aguina-Kang et al., 2024; Tam et al., 2024), 2D/3D layout for spatial constraint (Bahmani et al., 2023; Eldesokey & Wonka, 2024; Feng et al., 2024; Po & Wetzstein, 2024; Maillard et al., 2024; Yang et al., 2024), and scene graphs as spatial or semantic priors (Lin & Mu, 2024; Zhai et al., 2024; Gao et al., 2024). Chen et al. (2024a) generates objects from an image by decomposing the subscene into objects to generate separately. Alam & Ahmed (2024) propose to generate parametric CAD command sequences conditioned on an image. Our work follows the trend of conditional generation on structured data. To the best of our knowledge, we are the first to explore articulated object generation from a single image.

**Articulated object creation.** While more datasets have been introduced for articulated objects recently (Xiang et al., 2020; Liu et al., 2022; Geng et al., 2023), these datasets do not reach the scale and diversity of real-world objects, and still require manual annotation. To alleviate data scarcity, some works propose the creation of articulated objects from different input modalities. A-SDF (Mu et al., 2021) first learns to generate objects that can be deformed to different articulated states from a signed distance field. Then a line of work reconstructs articulated parts from point clouds collected at multiple states of the object (Jiang et al., 2022; Liu et al., 2023b), or from stereo-view (Heppert et al., 2023) or multi-view images observing the object in an open state (Mandi et al., 2024) or multiple states (Liu et al., 2023a; Weng et al., 2024), or from a video observing the object in motion (Song et al., 2024; Kerr et al., 2024). Generative models for articulated objects were recently proposed by NAP (Lei et al., 2023) and extended to a more controllable setting by CAGE (Liu et al., 2024c) to generate objects via specifying the part connectivity graph and object category. However, these prior works either require a laborious data collection process or lack intuitive control for generation. Our work aims to take a single image as input to create articulated objects, which is much cheaper and more practical for real-world applications. The work that shares the most similar setting with us is URDFormer (Chen et al., 2024b), which proposes a regression-based model to predict assets of articulated objects or a scene from an image. Considering the ambiguity posed by the image of an object in the resting state we propose to address the problem in a generative manner. This allows us to generate multiple plausible articulated objects from a single image while achieving better consistency to the input.

## 3 METHODS

### 3.1 OVERVIEW

Given an image $\mathcal{I}$ of an object in the resting (i.e. unopened) state, our goal is to generate a set of articulated parts $\{p_i\}_{i=1}^N$ assembling into a 3D object that is geometrically consistent with the input image while being kinematically plausible. To achieve this goal, we decompose the problem into two subtasks: 1) using the input image to guide the abstract-level generation of each part, and 2) fitting part meshes to the generated part arrangement and constructing them into a 3D articulated object.

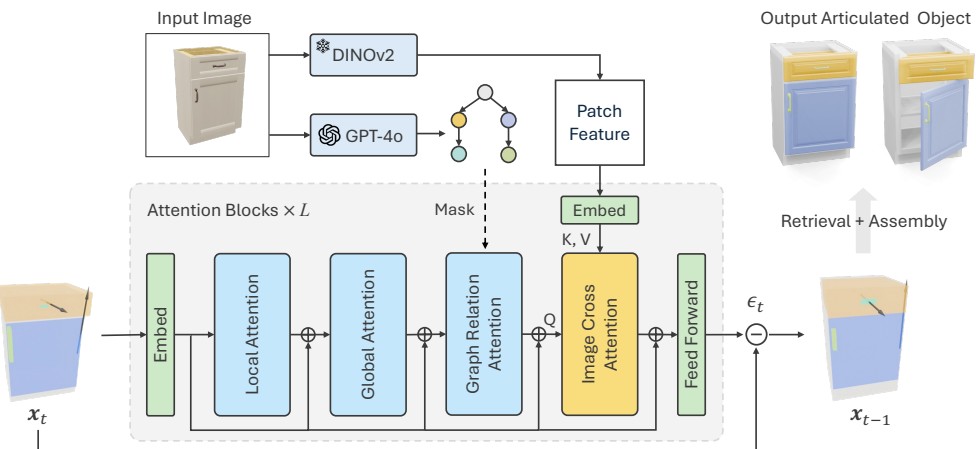

Figure 2: Our method takes an object image as input and generates attributes of articulated parts, which are used to assemble the object via part mesh retrieval. We design a diffusion-based model for part generation, which is guided by a part connectivity graph and DINOv2 patch features of the input image. Our denoising network is built on layers of attention blocks. The graph constraint is injected into the graph relation module by converting to an adjacency matrix as the attention mask. The image features act as the keys and values in the cross attention to condition the part arrangement.

To create the 3D object, one can generate each part's geometry or retrieve it from a predefined part library. As many parts of household objects are often geometrically simple, we choose to retrieve part geometries from a library. This works well in practice to assemble coherent objects, as shown in our experiments and also in previous works (Lei et al., 2023; Liu et al., 2024c).

In this work, we focus on the first subtask and we visualize the pipeline in Figure 2. We propose a diffusion-based model $\phi$ to learn a distribution of objects with articulated parts that resemble the input image. This distribution of objects visually matches the input image, but can vary in the part geometry and articulation to capture the ambiguity posed by the single snapshot of the object in the resting state that sometimes can be interpreted in multiple ways. Examples are illustrated in Figure 1. For each input image, our model learns to generate a list of attributes to describe each part in shape and motion. During generation, a part connectivity graph $G$ is also required as a guidance to determine the ordering of the list so that the parts are organized into a coherent articulated object. Thus, the model is essentially learning a distribution $p_\phi \left( \mathbf{x} : \{\mathbf{p}_i\}_{i=1}^N | \mathcal{I}, G \right)$. We will further explain the details of the data representation in Section 3.2 and the model design in Section 3.3.

## 3.2 DATA REPRESENTATION

We assume objects to be normalized inside a cube of size 2 centered at the origin and all the parameters are defined in the world coordinate system. We describe each articulated part with multiple attributes $\mathbf{p}_i = \{\mathbf{b}_i, l_i, t_i, \mathbf{a}_i, \mathbf{r}_i\}$ at abstract level, including the part bounding box $\mathbf{b}_i \in \mathbb{R}^6$, semantic label $l_i$, articulation type $t_i$, location and direction of joint axis $\mathbf{a}_i \in \mathbb{R}^6$, and motion range $\mathbf{r}_i \in \mathbb{R}^2$. The articulation type considered in this work includes *fixed*, *revolute*, *prismatic*, *continuous* and *screw* joints. The semantic label includes *base*, *door*, *drawer*, *handle*, *knob*, *tray*. The semantic label will be used to retrieve the part geometry from the dataset. Spatially, the parts are organized into a connectivity graph, where each node represents a part, and the edge direction indicates the parent-child relationship. This spatial relationship also defines the kinematic constraints between parts. During training, the graph is represented as an adjacency matrix $A \in \{0, 1\}^{N \times N}$, where $A_{ij} = 1$ if part $i$ is in connection with part $j$. The matrix $A$ will serve as an attention mask to guide the part generation process. To handle a different number of parts in the denoising diffusion process, we pad the parts to a maximum number 32 with all part attributes represented in 6 dimensions.

## 3.3 DENOISING NETWORK

### 3.3.1 IMAGE AND GRAPH GUIDANCE

**Image encoding.** To leverage images for generation guidance, we expect to encode the image into a latent feature that 1) is 3D-aware to capture the spatial orientation of the object; 2) is fine-grained

to detail the individual part semantically; and 3) contains the spatial information to infer the spatial structure and relationships between parts. We opt to use DINOv2 (Oquab et al., 2023), a ViT model pre-trained on a large-scale dataset and that has been shown with rich semantics and strong 3D awareness (El Banani et al., 2024), to encode our input image into latent patch features. Practically, we use the ViT-B/14 model with the register version (Darcet et al., 2023), which results in a total of 256 patches. These patch features are fed into our transformer-based denoising network as the conditioning input, where each patch acts as a token to be attended to. Our model learns to distill essential information from DINO features to understand the part semantic and spatial relationships.

**Part connectivity graph.** This graph is required to guide the part generation process. During training, we use the ground-truth graph provided in the dataset. For inference, we extract the graph from the input image using large vision-language models, which have been shown advance in visual reasoning tasks, such as image captioning and visual question answering. We empirically find that the connectivity graph of articulated parts can be effectively predicted from the image by leveraging the GPT-4o models with in-context learning. We prompt GPT-4o to take an object image as input, recognize all the articulated parts in the object, and then describe the part connectivity. Along with the instruction, we provide the model with examples of input-response pairs to help the model understand the context of the task. See the prompt in the appendix for more details.

### 3.3.2 MODEL ARCHITECTURE

We design our denoising network as a transformer-based architecture, which is shared for all timesteps. For each step, the model takes the output from the previous timestep $\mathbf{x}_{t-1}$ with the current timestep $t$ as input and produces a residual noise $\epsilon_t$ to be subtracted from $\mathbf{x}_{t-1}$ to construct $\mathbf{x}_t$ for the next iteration. The network consists of $L = 8$ layers in total, each of which has a layer normalization followed by attention modules with skip connections.

**Attention modules.** Each part attribute is regarded as a separate token during the attention process. We select CAGE (Liu et al., 2024c) as our base model that equips each attention block with: a *local attention* that harmonizes the attributes within each part; a *global attention* that coordinates parts globally to form a coherent object; and a *graph relation attention* that incorporates awareness of part connectivity from the graph. These self-attentions ensure the model to generate parts that adhere to the connectivity graph. To enforce geometric consistency with the input image, we further introduce an *image cross attention* at the end of each block; see Figure 2. Since the bounding-boxes directly reflect the shape and spatial locations of parts in the image, we only let the bounding-box parameters as queries to attend to image patches. Other attributes are then made compatible with the bounding box during the self-attention process followed. During cross-attention, we observe from the attention maps (see Figure 3) that each part is learned to focus on its relevant patches, indicating that each part is anchoring a specific region of the image during generation. This part-patch correspondence is learned without any explicit supervision during training, eliminating the need for taking part detection or segmentation as input.

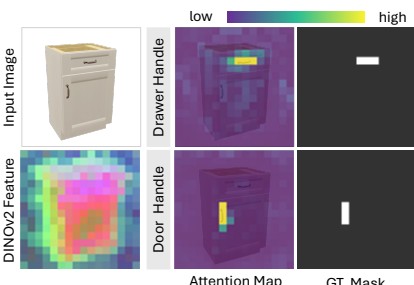

Figure 3: Attention maps for two example parts visualized at the 2nd last layer.

**Classifier-free training for conditioning.** Our denoising network takes image $\mathcal{I}$ as the main control and graph $G$ as auxiliary guidance. Both of them are critical as the image describes the spatial arrangement of parts and the graph determines the ordering of the parts to be generated so that the parts are organized correctly when articulating. Inherent from the base model, our network can also take object category $c$ as an additional condition, which is fused with the timestep embeddings via an adaptive norm layer (Xu et al., 2019) prior to attentions in each block. We empirically find that the original design of CAGE that takes them as hard constraints will lead the generation to be biased towards a narrow distribution of objects and overfit to the training data. It causes limited generalization to unseen images. To alleviate this issue, we propose to train our model in a classifier-free manner by dropping $c$ and $G$ randomly at a $50\%$ rate during training. In this way, the model is forced to generate objects that are visually consistent with the input image without being strictly constrained by them. Then $c$ can be regarded as an optional condition and $G$ as a

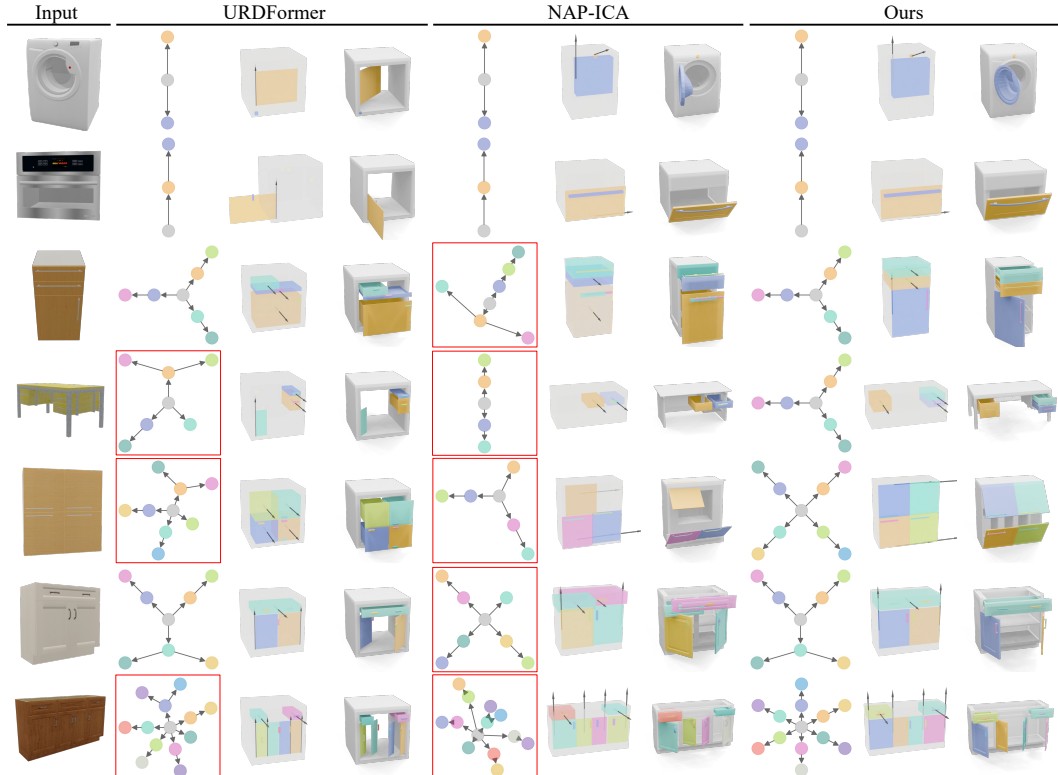

Figure 4: Qualitative comparison on the PartNet-Mobility test set. For each set of results, the first column shows the predicted part connectivity graph (the wrong ones are denoted in red boxes), the second column shows the part arrangement and joint for each part for the object in the resting state where the part coloring corresponds to the node in the graph, and the third column shows the final assets of the articulated object. Our method outperforms the baselines with better graph prediction, more consistent part arrangement with the input image, and more plausible part articulations.

condition secondary to the image input. At the same time, we also apply classifier-free training to $\mathcal{I}$ at $10\%$ dropping rate so that the image guidance can be further enhanced at inference by $\epsilon_\phi(\mathbf{x}_t; \mathcal{I}, G) = (1 + \omega)\epsilon_\phi(\mathbf{x}_t; t, \mathcal{I}, G) - \omega\epsilon_\phi(\mathbf{x}_t; t, \varnothing, G)$, where $\omega$ is a hyperparameter to balance the influence of the image and the graph, and $\varnothing$ denotes the absence of the image.

### 3.3.3 TRAINING OBJECTIVES

We supervise our model with the loss $\mathcal{L} = \mathcal{L}_\epsilon + \lambda\mathcal{L}_{\text{fg}}$, where $\mathcal{L}_\epsilon$ is noise residual loss and $\mathcal{L}_{\text{fg}}$ is object foreground loss. Our model follows the training procedure of the denoising diffusion probabilistic model (Ho et al., 2020), and $\mathcal{L}_\epsilon$ is the standard diffusion loss, which computes the mean squared error between the output $\epsilon_t$ and the ground truth residual noise $\hat{\epsilon}_t$ that is added to the input $\mathbf{x}_{t-1}$ in the timestep $t$ in the diffusion process. $\mathcal{L}_{\text{fg}}$ is an auxiliary loss that helps focus the attention on the foreground patches of the object region for all parts and all layers. The loss $\mathcal{L}_{\text{fg}}$ supervises the cross attention map $\mathcal{A}^l_{\mathbf{p}_i}$ for part $\mathbf{p}_i$ at layer $l$, which records the attention scores from the part to all the image patches. Formally, we define the loss as $\mathcal{L}_{\text{fg}} = \frac{1}{L}\sum_{l=1}^{L}\sum_{i=1}^{N}(1 - \mathcal{A}^l_{\mathbf{p}_i} \circ \mathcal{M}_{\text{fg}} + \mathcal{A}^l_{\mathbf{p}_i} \circ \neg\mathcal{M}_{\text{fg}})$, where $\mathcal{M}_{fg}$ is the foreground mask of the object projected onto the patches, and $\circ$ denotes the element-wise product. We use $\lambda = 0.1$ to balance the two losses.

## 4 EXPERIMENTS

### 4.1 DATASET

We collect data from PartNet-Mobility dataset (Xiang et al., 2020) to train our model across 7 categories (i.e., `Storage`, `Table`, `Refrigerator`, `Dishwasher`, `Oven`, `Washer`, and

Table 1: Evaluation of the reconstruction quality on the PartNet-Mobility test set. We report the results under oracle setting with ground-truth part detection given to URDFormer and ground-truth graph given to NAP-ICA and ours as a highline reference in the upper part of the table. The lower part shows the results on the real setting that only an image is given, and all the methods produce one object per input. Our method outperforms other baselines in both oracle and real settings.

| | Reconstruction quality | | | | | | Collision | Graph |
|---|---|---|---|---|---|---|---|---|
| | RS-$d_{\text{gIoU}}$ ↓ | AS-$d_{\text{gIoU}}$ ↓ | RS-$d_{\text{cDist}}$ ↓ | AS-$d_{\text{cDist}}$ ↓ | RS-$d_{\text{CD}}$ ↓ | AS-$d_{\text{CD}}$ ↓ | AOR↓ | Acc% ↑ |
| URDFormer-GTbbox | 1.0861 | 1.0882 | 0.1471 | 0.3225 | 0.3400 | 0.6031 | 0.0616 | 83.55 |
| NAP-ICA-GTgraph | 0.6830 | 0.6915 | 0.0739 | 0.2206 | 0.0282 | 0.2646 | 0.0105 | 44.81 |
| Ours-GTgraph | **0.4381** | **0.4541** | **0.0315** | **0.0751** | **0.0089** | **0.0836** | **0.0014** | **100.0** |
| URDFormer | 1.1868 | 1.1879 | 0.2693 | 0.4535 | 0.5502 | 0.8374 | 0.2341 | 32.03 |
| NAP-ICA | 0.5778 | 0.5854 | 0.0501 | 0.0979 | 0.0173 | 0.0914 | 0.0120 | 75.97 |
| Ours | **0.4841** | **0.4974** | **0.0448** | **0.0955** | **0.0168** | **0.0905** | **0.0043** | **82.47** |

`Microwave`). We render 20 images per object from random viewpoints, sampled within a 90° horizontal range and a 90° upwards vertical range relative to the object's front. With several augmentation strategies applied, we end up with 3,063 objects paired with 20 images rendered at resting state for training, and additional 77 objects paired with 2 random views for testing. We also use 135 objects from the ACD dataset (Iliash et al., 2024) for additional evaluation in the zero-shot manner to test the generalization capability. In total, we have 55K training samples and 424 test samples in the experiments. Please see the appendix for more details on data augmentation.

## 4.2 BASELINES

We compare our method with two baselines: URDFormer (Chen et al., 2024b) as the state-of-the-art that works in the same setting and NAP (Lei et al., 2023) as representative prior work that unconditionally generates articulated objects. For URDFormer, we take their pretrained model for part detection on the input image and re-implement their object-level module to finetune on our training data for articulated object prediction. For NAP, we re-implement their method by plugging our image cross-attention module to attend with their node attributes after each of their graph attention layer to enable NAP to condition on an image. This variant of NAP is denoted by *NAP-ICA*. We also replace the shape latent originally used in NAP with part semantic label for sharing our retrieval algorithm, which is a lightweight version of NAP that has been shown to perform better in a conditional setting by Liu et al. (2024c). For comparison, all the methods are trained with or finetuned on the same training set and evaluated on the same test set. The final geometry of objects is from retrieval across all baseline methods. Please see the appendix for more implementation details.

## 4.3 METRICS

To evaluate the reconstruction accuracy in terms of shape structure and kinematics, we adopt 3 sets of evaluation metrics to measure the similarity between the output and the ground-truth object that corresponds to the given image. For each metric set, the object similarity is measured both in the resting state (**RS-**) and across the articulated states (**AS-**) uniformly sampling from resting to the end state. The overall object scale is normalized to the one of the ground-truth objects to account for scale ambiguity across methods. Each metric is reported by averaging over the pairwise distance per part per state, and the part pairing is determined by the Hungarian algorithm.

To formalize all the evaluation metrics, we denote $p_i$ as the $i$-th part in one object $O_1$ and $q_j$ as the $j$-th part in the other object $O_2$. The pairwise distance between $p_i$ and $q_j$ is calculated as $d^s(p_i, q_j)$ in state $s$ among all sampling states $S$. The Hungarian matching algorithm $\mathcal{H}_{O_1 \to O_2}(p_i)$ takes the centroid distance of the part as the cost matrix to determine the pairing of parts from $O_1$ to $O_2$ and returns the corresponding part for each part $p_i$ in $O_1$. Our metrics $\mathcal{D}$ are computed as follows.

$$\mathcal{D} = \frac{1}{|S|} \sum_{s \in S} \left( \frac{1}{2} \left( \frac{1}{N} \sum_{i=1}^{N} d^s\left(p_i, \mathcal{H}_{O_1 \to O_2}(p_i)\right) + \frac{1}{M} \sum_{j=1}^{M} d^s\left(q_j, \mathcal{H}_{O_2 \to O_1}(q_j)\right) \right) \right) \quad (1)$$

The three metrics we adopt to compute the pairwise distance $d^s(p_i, q_j)$ are as follows.

- $d_{\text{gIoU}}$ ↓: At the abstract level of the part description, we report the distance based on generalized intersection over union (gIoU) (Rezatofighi et al., 2019) of the bounding box volume for each pair of parts as $1 - \text{gIoU}$, lower-bounded by 0.

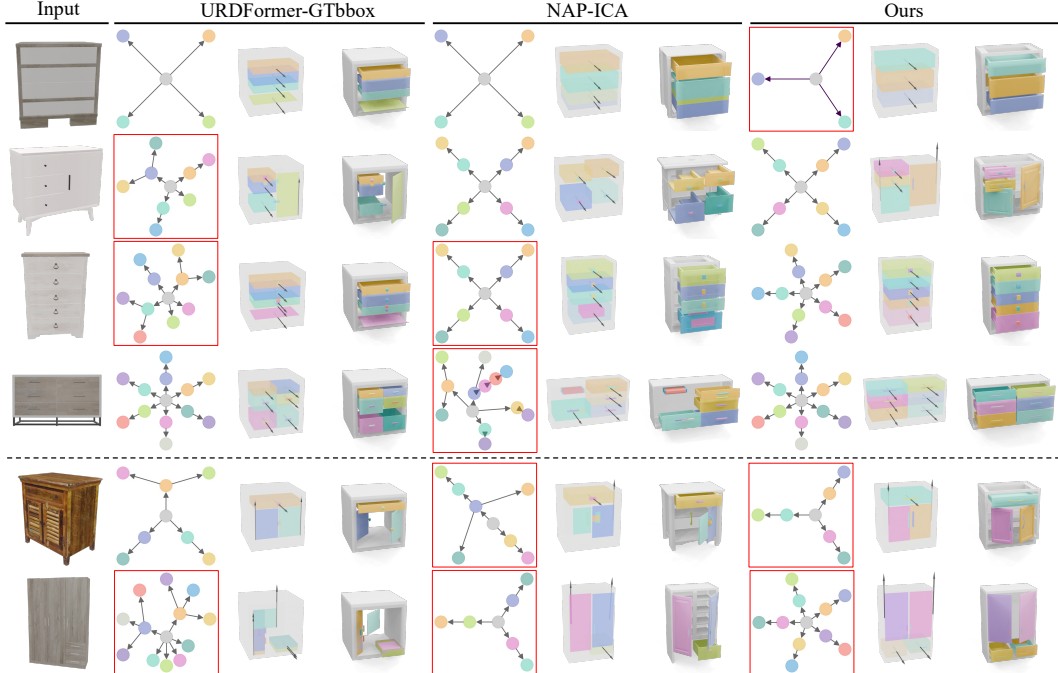

Figure 5: Qualitative comparison on the ACD dataset in a zero-shot testing. The first four rows show that our method can generate more geometrically accurate objects with plausible motions compared to the baselines. The red boxes denote incorrect part connectivity graphs relative to the ground truth. The first row shows that even when our predicted graph is different from the ground truth due to ambiguity in the image, our method can still generate a realistic and reasonable object. The last two rows show failure cases for two challenging input images. When the texture is complex or the part arrangement is cluttered, our method may not accurately recover some details (e.g., two knobs on drawer merged into one handle in $2^{nd}$ to last example; doors and drawers misplaced in last example).

- $d_{\textbf{cDist}} \downarrow$: Since small and thin parts can easily miss the overlapping region, we also report the Euclidean distance between the centroids of two parts, lower bounded by 0. We find it to reflect the part-level similarity in a more fine-grained manner than gIoU.
- $d_{\textbf{CD}} \downarrow$: At the mesh level, we report the Chamfer Distance averaged over the 2,048 points sampled on the mesh surface per part and computed in symmetric, lower-bounded by 0.

Our formulation of $\mathcal{D}$ is built upon the metrics Instantiation Distance (ID) introduced by NAP and Abstract Instantiation Distance (AID) proposed by CAGE. Initially, ID only accounts for CD over the entire object, minimizing across all possible articulated states while neglecting part-level similarity and state matching. AID addresses this by incorporating state pairing and part matching to compute IoU, but part pairing is determined using a greedy algorithm, which may not yield optimal results. In our refined metrics, we employ the Hungarian algorithm for optimal part pairing and extend the pairwise distance to three variations, capturing multiple aspects of part-level similarity more effectively. Please see the appendix for more details of the prior metrics.

To evaluate the accuracy of the graph inferred from the image, we report the accuracy (**Acc**% ↑) in terms of the percentage of correct graph topology. We also report the average overlapping ratio (**AOR**↓) introduced by CAGE to detect unrealistic collisions among sibling parts in the graph.

## 4.4 RESULTS

**Reconstruction quality.** We show the quantitative comparison on PartNet-Mobility dataset in Table 1 and qualitative results in Figure 4. In terms of the ***topological correctness of the predicted graph***, our GPT-4o module achieves the highest accuracy. The graph prediction of URDFormer is less accurate, and we observe that it is largely bottlenecked by their part detection module which often misses parts. NAP-ICA predicts more accurate graphs when node connectivity is not enforced as a constraint, which is consistent with the observation in CAGE that NAP struggles to adhere to

Table 2: Evaluation of the reconstruction quality on the ACD testset. We report the results under oracle setting with ground-truth part detection given to URDFormer and ground-truth graph given to NAP-ICA and ours as a highline reference in the upper part of the table. The lower part shows the results in the real setting where only an image is given, and all the methods produce one object per input. Our method outperforms other baselines in better generalization to more complex data.

| | Reconstruction quality | | | | | | Collision | Graph |
|---|---|---|---|---|---|---|---|---|
| | RS-$d_{\text{gIoU}} \downarrow$ | AS-$d_{\text{gIoU}} \downarrow$ | RS-$d_{\text{cDist}} \downarrow$ | AS-$d_{\text{cDist}} \downarrow$ | RS-$d_{\text{CD}} \downarrow$ | AS-$d_{\text{CD}} \downarrow$ | AOR$\downarrow$ | Acc% $\uparrow$ |
| URDFormer-GTbbox | 1.1986 | 1.2012 | 0.2292 | 0.2931 | 0.4343 | 0.4965 | 0.1209 | 48.53 |
| NAP-ICA-GTgraph | 1.0585 | 1.0631 | 0.1987 | 0.2810 | 0.1376 | 0.2417 | 0.0193 | 36.67 |
| Ours-GTgraph | **0.9842** | **0.9881** | **0.1620** | **0.2024** | **0.1171** | **0.1784** | **0.0112** | **100.0** |
| URDFormer | 1.2288 | 1.2309 | 0.2914 | 0.4285 | 0.7198 | 0.8995 | 0.2840 | 4.48 |
| NAP-ICA | 1.0233 | 1.0286 | 0.1691 | 0.2331 | 0.1110 | 0.1887 | 0.0133 | 16.67 |
| Ours | **0.9517** | **0.9574** | **0.1526** | **0.1983** | **0.1011** | **0.1679** | **0.0085** | **39.34** |

graph conditioning. In terms of the ***geometric consistency with the input image***, we observe that our method and NAP-ICA handle more variational input views and have better part-level understanding from the image than URDFormer, which leads to higher reconstruction quality by referring to the "RS-" metrics and qualitative results in Figure 4. This high performance is largely attributed to the ICA module we proposed that enables us and NAP to efficiently distill part knowledge from the rich DINO features. Our method further outperforms NAP-ICA, which is benefiting from the efficient graph guidance to better coordinate parts, especially when the number of parts gets larger. In terms of the ***plausibility of the part articulation***, our method can generate more kinematically plausible objects with less part collisions by referring to the "AS-" and AOR metrics. It is important to note that the 'AS-' metrics cannot fully capture motion plausibility, as there are cases where the ground-truth motion is not unique. From Figure 4, we observe that URDFormer exhibits inaccurate joint axis of doors due to its reliance on a predicted label that conflates semantic and joint categorizations, thereby impairing joint parameter estimation in their post-processing step. NAP-ICA tends to produce more collisions and less plausible motions, which is also reflected in the user study below.

**Overall realism evaluated through user study.** To further evaluate the realism of the reconstructed objects, we conduct a user study in terms of how realistic the generated objects refer to the input images. In our user study, participants were presented with three output objects given the same input image from three methods each time, and 15 pairs of results in total were shown. The task for each participant was to rank these objects based on their perceived realism, specifically evaluating how closely each object aligned with the input image in terms of visual consistency and how plausible the object articulation is. We collect responses from 48 participants and report the averaged ranking of each method ranging from 1 as the best to 3 as the worst. As a result, our method ranks the highest with an average of 1.20, followed by NAP-ICA with 1.91 and URDFormer with 2.89.

**Generalization to unseen dataset.** To further evaluate the generalization capability, we show the comparison results on the ACD dataset in a zero-shot manner. The ACD dataset comprises objects with greater structural complexity, geometric diversity, and appearance variances compared to the PartNet-Mobility dataset. Figure 5 shows examples from the ACD dataset, where our method generalize better to objects with more challenging textures and more complex part structures that are out-of-distribution. As shown in Table 2, we outperform other baselines overall by a large margin.

**Comparison with multi-view reconstruction.** We compare with Real2Code (Mandi et al., 2024), a state-of-the-art method for articulated object reconstruction from multi-views. As their code is not fully open-sourced, we follow the evaluation protocol shown in their paper. Specifically, Real2Code reconstructs an object from 12 views observing the object in an opening state, which reveals much more geometric

Table 3: Comparison with Real2Code. The CD $\times 10^3$ is reported for the whole object and parts on average.

| Category | Fridge | | Furniture | | Furniture | |
|---|---|---|---|---|---|---|
| Number of Parts | 2-3 | | 2-4 | | 5-15 | |
| Metric | whole | part | whole | part | whole | part |
| Real2Code-GTseg | **0.51** | **2.04** | **1.46** | **3.30** | 5.84 | 16.80 |
| Ours-GTgraph | 4.06 | 2.78 | 13.77 | 9.86 | **5.23** | **8.54** |
| Real2Code | **0.60** | **1.28** | **3.47** | 65.79 | 19.70 | 118.58 |
| Ours | 6.67 | 6.35 | 14.29 | **13.69** | **11.81** | **28.67** |

and kinematic information than our setting that only observes the object in a resting state from a single view. We retrain our model on their data split with our augmented data and evaluate on their test set to ensure a fair comparison. From Table 3, we observe that our method achieves competitive results on simple objects with less parts and can even outperform Real2Code on objects with more parts in complex structures. Also, our score gap between CD on the whole and its part on average is smaller than Real2Code, indicating that our method reconstructs better part geometry.

Table 4: The ablation study of the design components of our method on top of the base model, in terms of the image cross-attention (ICA) module, foreground loss $\mathcal{L}_{\text{fg}}$, and the classifier-free training on the object category (cCF), graph constraint (gCF), and image guidance (iCF). The average score of 5 samples per input given GT part connectivity graph is reported on the ACD dataset.

| | RS-$d_{\text{gIoU}}$ ↓ | AS-$d_{\text{gIoU}}$ ↓ | RS-$d_{\text{cDist}}$ ↓ | AS-$d_{\text{cDist}}$ ↓ | RS-$d_{\text{CD}}$ ↓ | AS-$d_{\text{CD}}$ ↓ |
|---|---|---|---|---|---|---|
| Base | 1.2059 | 1.2094 | 0.2985 | 0.3593 | 0.3469 | 0.4705 |
| Base + ICA | 1.0399 | 1.0433 | 0.1991 | 0.2492 | 0.1459 | 0.2372 |
| Base + ICA + cCF | 1.0102 | 1.0141 | 0.1874 | 0.2312 | 0.1423 | 0.2111 |
| Base + ICA + cCF + $\mathcal{L}_{\text{fg}}$ | 1.0028 | 1.0067 | 0.1857 | 0.2253 | 0.1403 | 0.2009 |
| Base + ICA + cCF + $\mathcal{L}_{\text{fg}}$ + iCF | 0.9887 | 0.9973 | 0.1678 | 0.2109 | 0.1184 | 0.1853 |
| Base + ICA + cCF + $\mathcal{L}_{\text{fg}}$ + gCF + iCF (Ours) | **0.9528** | **0.9565** | **0.1530** | **0.1989** | **0.1014** | **0.1701** |

**Ablation studies.** We ablate the design components of our method as shown in Table 4, where the average of 5 samples per input given GT part connectivity graph is reported on the ACD dataset. On top of our base model, the ICA module plays the main role in enabling image conditioning by distilling DINOv2 features to part attributes via cross-attention. Its effectiveness is also demonstrated by adding this module to NAP and achieving performance close to our method. Removing object category (cCF) as a hard constraint allows the model to share knowledge across categories, which enhances the efficiency of training data utilization and leads to improved generalization. $\mathcal{L}_{\text{fg}}$ yields a modest improvement in performance by concentrating attention on the object region. The classifier-free training for the image and graph guidance enhances the model to the best performance by modulating the distribution further to respect the input image and generalize better to unseen data. We also ablate on how the augmented data affects the performance in the supplement, where the augmentation strategies are shown to be effective in improving the generalization capability.

**Failure cases and limitations.** Some failure cases are shown in Figure 5 in the last two rows. We observe three failure modes in our method: 1) incorrect graph predictions from input images with challenging textures; 2) less accurate part geometry, particularly for small parts (e.g., knobs in the second last example); and 3) inaccurate part arrangement, often due to missing components in the predicted graph or when the object's structure is highly cluttered and complex (see the last example). These failures are due to the low resolution of the input image that provides limited part details for our model to infer accurate part geometry and arrangement, especially for complex objects with a large number of parts compacted in a small number of patch features. The unbalanced training on the data with less complex structure and simple textures also hinder the generalization ability of our model. Further augmentation to balance the object complexity and increase texture diversity will improve generalization to real settings for future work.

## 5 CONCLUSION AND FUTURE WORK

We address the problem of reconstructing household articulated objects from a single image in a generative manner. We break down this complex task into sub-tasks solving the problem from coarse to fine: we first infer part connectivity into a graph from the image, and then use the extracted graph and input image as guidance to produce part arrangement and motion parameters to describe the parts abstractly, and we finally retrieve the mesh for each part to assemble the full object. We develop a diffusion model with attention mechanisms to distill DINOv2 features of the image and to leverage graphs together to condition the part generation. We demonstrate the effectiveness and generalizability of our method on PartNet-Mobility and ACD datasets, and show that our method outperforms the state-of-the-art methods.

While our work takes important first steps on this challenging task, several aspects remain open for further exploration. Currently, our method is tailored to specific object categories, primarily those with cuboid shapes. Extending this approach to handle a wider range of object categories with more complex geometries is interesting for future research. From a geometric perspective, although our mesh retrieval algorithm performs well for cabinet-like objects, it falls short in capturing finer geometric details of parts as seen in the input image. A promising avenue for future work lies in developing methods that generate part geometries more faithfully aligned with the input image, enhancing both the accuracy and realism of the reconstruction. We hope this work encourages the community to explore these directions and push the boundaries of articulated object reconstruction.

ACKNOWLEDGMENTS

This work was funded in part by a CIFAR AI Chair, a Canada Research Chair, and NSERC Discovery grants, and enabled by support from the Digital Research Alliance of Canada. We thank Han-Hung Lee, Hou In Ivan Tam, Qirui Wu, and Yiming Zhang for helpful discussions.

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

# A APPENDIX

## A.1 IMPLEMENTATION DETAILS

**Our model training.** We empirically find that initializing our base model with the CAGE pretrained weights helps our model converge faster. The intuition is that the pretrained base model has already learned a good distribution of articulated objects conditioned on the graph, and our model can further modulate the distribution to respect the input image. We train our model for 200 epochs after initializing CAGE pretrained weights with a batch size of 40 and each with 16 timesteps sampled from the diffusion process for each iteration. We train 1,000 diffusion steps in total. We use the AdamW Loshchilov (2017) optimizer with learning rate $5e - 4$ for ICA module and $5e - 5$ for the base model parameters, and the beta values are set to $(0.9, 0.99)$. We schedule the learning rate with 3 epochs of warm-up from $1e - 6$ to the initial learning rate and then consine annealing to $1e - 5$. The network has 6 layers of attention blocks with 4 heads and 128 hidden units. Our model is trained on 4 NVIDIA A40 GPUs for 23 hours.

**NAP-ICA training.** In Figure 6, we show how we integrate the ICA module into the NAP (Lei et al., 2023) framework. In between each graph layer used in NAP for node and edge feature fusion, we add an ICA module to condition the node attributes on the image patch feature distilled from DINOv2 (Oquab et al., 2023). Once updated the node attributes, we concatenate the node and edge features to pass through the next graph layer. For node attributes, NAP originally uses a shape latent code to encode the part geometry in 128-dimensional vector. In this work, we replace it with a semantic label, which is shown in CAGE (Liu et al., 2024c) that this

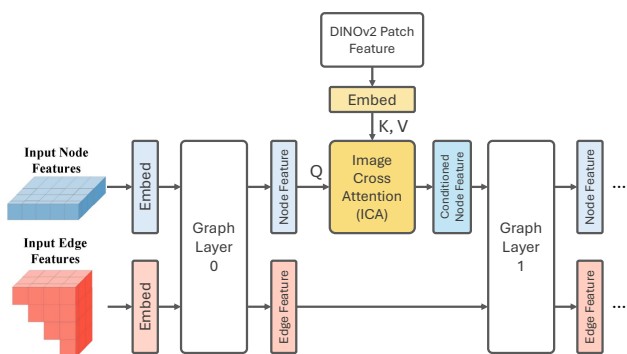

Figure 6: Network architecture of NAP-ICA. We integrate our ICA module into the NAP framework by adding it in between the graph layers and let the node attributes to be conditioned on the image feature distilled from DINOv2.

lightweight version of NAP performs better in conditional setting as it balances out with other attributes in latent space. We follow this simplification in our implementation with another benefit of reducing the model's capacity of cross-attention with image patches. And we show in the experiments that this light version of NAP works surprisingly well in our task by incorporating our ICA module. We adopt a similar training strategy for NAP-ICA as above by using a pre-trained NAP to initialize the network parameters except for the ICA module, otherwise the training takes much longer to converge. We use the AdamW optimizer with learning rate $1e - 4$ for all parameters, and other hyperparameters are the same as the original NAP model to train NAP-ICA in 100 epochs on a NVIDIA A40 GPU for 25 hours.

**URDFormer training.** As URDFormer (Chen et al., 2024b) has not released the training code, we tried our best to re-implement their object module to match the original implementation by referring to the paper and asking the authors for details. Additionally, we adapt the architecture to match the correct number of part categories, as originally *doorU* (door that opens upwards) is missing from the architecture and checkpoint. We take their pre-trained weights and fine-tune the network on our training data for 100 epochs with learning rate $1e - 5$ and batch size 256. We find that the loss weights provided by authors, connectivity loss weighted by 5 and other losses by 1, do not work well in our case. Therefore, we select the weights that lead to convergence - 20 for connectivity loss and 4 for base part type loss. Additionally, we find that gradient clipping is detrimental and therefore train without it.

**Mesh retrieval algorithm.** Our mesh retrieval algorithm is adapted from the implementation of CAGE (Liu et al., 2024c). They designs to first determine the most similar objects as candidate by computing the Abstract Instantiation Distance (AID) score between the generated object and the objects in the database. This step helps to find the best base part for the generated object as they assume that the AID score reflects the similarity of articulation of all parts in the object. We

borrow this idea and follow this step with one change that we use our AS-$d_{cDist}$ score to determine the candidate objects. Then we use the part semantic label to match the part geometry from the candidate object for each part. If there are multiple parts with the same semantic label, we re-use the mesh for these parts to ensure a style consistency in the generated object. Our retrieval algorithm is also shared with NAP-ICA model to assemble the final object.

## A.2 PART CONNECTIVITY GRAPH PREDICTION USING GPT-4O

We empirically find that GPT-4o performs well in understanding articulated part relationships and predicting part connectivity as a graph by taking an object image as input. We choose the version `gpt-4o-2024-08-06` for our experiments. We provide GPT-4o with step-by-step instructions to teach the model to understand the articulated part in the image and then connect them into a graph. Here is our instruction:

---

**System:**

You are an expert in the recognition of articulated parts of an object in an image.

You will be provided with an image of an articulated object. You should follow the following steps to achieve the task:

- 1) Recognize all the articulated parts of the object in the image, in the choice of ['base', 'door', 'knob', 'handle', 'drawer', 'tray']. Note that there should be always one "base" part, and trays can only exist in microwaves. Each handle or knob should be attached to a door or a drawer. Each door should only have one handle or knob at most.

- 2) Describe how the parts are connected and then organize them in a part connectivity graph. The "base" part is always the root of the graph.

Here is an example of your response:

I recognize all the articulated parts of a storage furniture, they are:

base, drawer (attach to base), handle (attach to drawer), handle (attach to drawer), door (attach to base), handle (attach to door).

The part connectivity graph for the object is:
```json
{"base": [{"drawer": [{"handle": []}, {"handle": []}]}, {"door": [{"handle": []}]}]}
```

---

Figure 7: Prompt for GPT-4o to predict the part connectivity graph in the object image.

Apart from the instruction, we also provide the model with several examples of conversational dialogues, which is critical for GPT-4o to scope the context and understand what the user is asking for. We find that the example input as a description of the image outperforms the example input as a real image, and here is an example of the conversation:

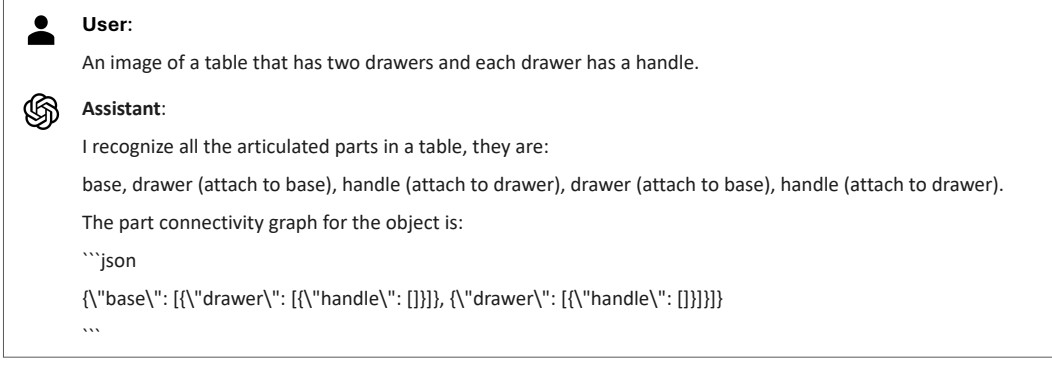

Figure 8: A conversation example to prompt GPT-4o for better context understanding.

Table 5: The ablation of the effectiveness of our augmented data (aug) by testing on the PartNet-Mobility dataset (PM) and ACD dataset (ACD). The scores in average of 5 samples per input given GT part connectivity graph are reported show consistent trend of the performance.

| Testing data | Training data | Reconstruction quality | | | | | | Collision |
| | | RS-$d_{\text{gIoU}}$ ↓ | AS-$d_{\text{gIoU}}$ ↓ | RS-$d_{\text{cDist}}$ ↓ | AS-$d_{\text{cDist}}$ ↓ | RS-$d_{\text{CD}}$ ↓ | AS-$d_{\text{CD}}$ ↓ | AOR↓ |
| --- | --- | --- | --- | --- | --- | --- | --- | --- |
| PM | w/o aug | 0.5965 | 0.6101 | 0.0827 | 0.1427 | 0.0606 | 0.1725 | 0.0032 |
| | w/ aug | **0.4414** | **0.4566** | **0.0309** | **0.0752** | **0.0097** | **0.0841** | **0.0027** |
| ACD | w/o aug | 1.1270 | 1.1306 | 0.2260 | 0.2782 | 0.1948 | 0.2783 | 0.0156 |
| | w/ aug | **0.9528** | **0.9565** | **0.1530** | **0.1989** | **0.1014** | **0.1701** | **0.0130** |

## A.3 DATASET DETAILS

**Data augmentation.** We design several augmentation strategies to enhance the diversity of the training data in terms of part arrangement and articulation: 1) parts randomization by swapping the handles/knobs across objects and perturbing their positions; 2) diversifying part arrangements by rotating the objects upside-down; 3) creating objects with more complex structures by stacking multiple objects together; 4) densifying the data distribution by randomizing the object scale. We carefully design heuristics to ensure the augmented data is still plausible and realistic, and the above strategies are only applied to `Table` and `StorageFurniture` categories.

**Ablations on the data augmentation.** We conduct ablation studies to investigate the effectiveness of augmentation strategy in performing our tasks on PartNet-Mobility dataset and generalization to ACD dataset. We show the quantitative results in Table 5 by reporting the average scores of 5 samples per input given ground-truth part connectivity graph. Reporting the average score of multiple samples helps to reduce the variance of the results and show consistent trend of the performance. We observe that our data augmentation significantly improves our performance on both datasets.

**Dataset statistics.** We provide the statistics of the data we use in our experiments from ACD dataset and the PartNet-Mobility dataset before (PM-ori) and after our data augmentation (PM-aug). We filter out the objects with broken part segmentation from the PartNet-Mobility dataset and the L-shaped tables and unrelated categories (e.g., *bed, trashcan, barbecue, bench, and safe*) from the ACD dataset.

Table 6: Statistics of the ACD and PartNet-Mobility dataset before (PM-ori) and after data augmentation (PM-aug). We report the number of objects in different shape complexity and in total for each dataset.

| | Number of Articulated Parts Per Object | | | | | | |
| Dataset | 0-5 | 5-10 | 10-15 | 15-20 | 20-25 | 25-32 | Total |
| --- | --- | --- | --- | --- | --- | --- | --- |
| ACD | 67 | 32 | 25 | 10 | 1 | 0 | 135 |
| PM-ori | 403 | 135 | 24 | 3 | 1 | 0 | 566 |
| PM-aug | 1,673 | 1,041 | 244 | 150 | 47 | 5 | 3,140 |

## A.4 EVALUATION METRICS

We proposed our evaluation metrics $\mathcal{D}$ in the main paper to better capture the part-level reconstruction quality than the existing metrics in the literature: Instantiation Distance (ID) proposed in NAP (Lei et al., 2023) and Abstract Instantiation Distance (AID) proposed in CAGE (Liu et al., 2024c). In Figure 9, we use two examples to illustrate the limitation of these metrics. When evaluating the reconstruction quality of the generated object with respect to the ground-truth object, we observe that the ID and AID metrics may not reflect the quality of the reconstruction well and misalign with the human perception. The objects in the red box show that there are

Table 7: We report the reconstruction quality of our method and other baselines with the evaluation metrics **ID** proposed in NAP and **AID** proposed in CAGE.

| Testing Set | Methods | ID↓ | AID↓ |
| --- | --- | --- | --- |
| PM | URDFormer | 0.0784 | 0.7565 |
| | NAP-ICA | **0.0076** | 0.6224 |
| | Ours | 0.0085 | **0.5628** |
| ACD | URDFormer | 0.0765 | 0.7374 |
| | NAP-ICA | **0.0221** | 0.7553 |
| | Ours | 0.0229 | **0.6989** |

situations where a lower ID or AID score does not necessarily mean a better reconstruction quality. The reason is that ID doesn't consider either part-level matching nor the state-level matching between the objects, and AID matches the parts in a greedy way, which may lead to suboptimal part matching and inaccurate measurement of the reconstruction quality. While ID and AID have their own limitations in measuring the reconstruction quality, we still report the evaluation results using these metrics here for completeness. From Table 7, we achieve the best AID score on both datasets, which is consistent with the results reported in the main paper.

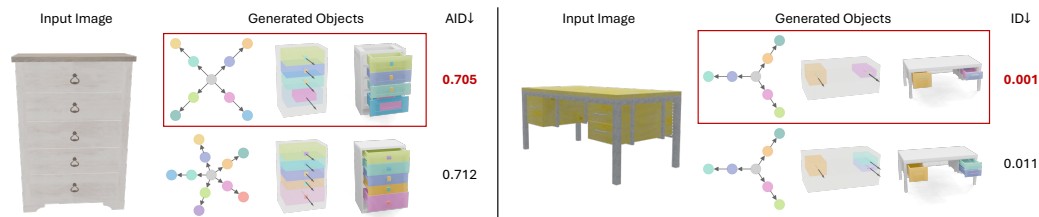

Figure 9: Examples illustrating the limitations of the AID and ID metrics in evaluating reconstruction quality, as they may not capture part-level accuracy and also misalign with human perception. The objects in the red box are examples with worse reconstruction quality but lower AID and ID.

## A.5 MORE EXPERIMENT RESULTS

**Reconstruction quality and generalization across multiple outputs.** We provide more quantitative results by reporting the average scores of 5 samples per input for NAP-ICA and our model in comparison with URDFormer on the PartNet-Mobility dataset in Table 8 and generalization on the ACD dataset in Table 9. The average scores better reflect the consistency of the output quality of the models compared to the single output reported in the main paper. From Table 8, we observe that our model consistently outperforms other baselines in terms of the reconstruction quality and collision rate between the parts. From Table 9, we observe that our model consistenly generalizes much better to the ACD dataset compared to other baselines.

Table 8: Evaluation of the reconstruction quality on the PartNet-Mobility testset. For NAP-ICA and ours, we report the average results across 5 outputs per input. Our lowest scores across all metrics indicate that our model can consistently produce high-quality outputs when generating multiple objects with different random seeds.

| | Reconstruction quality | | | | | | Collision | Graph |
|---|---|---|---|---|---|---|---|---|
| | RS-$d_{\text{gIoU}}$ ↓ | AS-$d_{\text{gIoU}}$ ↓ | RS-$d_{\text{cDist}}$ ↓ | AS-$d_{\text{cDist}}$ ↓ | RS-$d_{\text{CD}}$ ↓ | AS-$d_{\text{CD}}$ ↓ | AOR↓ | Acc% ↑ |
| URDFormer-GTbbox | 1.0861 | 1.0882 | 0.1471 | 0.3225 | 0.3400 | 0.6031 | 0.0616 | 83.55 |
| NAP-ICA-avg-GTgraph | 0.6834 | 0.6914 | 0.0732 | 0.2214 | 0.0287 | 0.2774 | 0.0134 | 42.59 |
| Ours-avg-GTgraph | **0.4693** | **0.4825** | **0.0316** | **0.0767** | **0.0097** | **0.0879** | **0.0050** | **100.0** |
| URDFormer | 1.1868 | 1.1879 | 0.2693 | 0.4535 | 0.5502 | 0.8374 | 0.2341 | 32.03 |
| NAP-ICA-avg | 0.5753 | 0.5854 | 0.0598 | 0.1072 | 0.0189 | 0.1074 | 0.0224 | 75.84 |
| Ours-avg | **0.5126** | **0.5240** | **0.0459** | **0.0975** | **0.0181** | **0.1032** | **0.0032** | **82.47** |

Table 9: Evaluation of the reconstruction quality on the ACD testset in a zero-shot manner. For NAP-ICA and ours, we report the average results across 5 outputs per input. Our lowest scores across all metrics indicate that our model can consistently produce high-quality outputs from different random seeds and generalize well to a more complex dataset.

| | Reconstruction quality | | | | | | Collision | Graph |
|---|---|---|---|---|---|---|---|---|
| | RS-$d_{\text{gIoU}}$ ↓ | AS-$d_{\text{gIoU}}$ ↓ | RS-$d_{\text{cDist}}$ ↓ | AS-$d_{\text{cDist}}$ ↓ | RS-$d_{\text{CD}}$ ↓ | AS-$d_{\text{CD}}$ ↓ | AOR↓ | Acc% ↑ |
| URDFormer-GTbbox | 1.1986 | 1.2012 | 0.2292 | 0.2931 | 0.4343 | 0.4965 | 0.1209 | 48.53 |
| NAP-ICA-avg-GTgraph | 1.0588 | 1.0646 | 0.1996 | 0.2841 | 0.1389 | 0.2542 | 0.0225 | 36.51 |
| Ours-avg-GTgraph | **0.9850** | **0.9891** | **0.1624** | **0.2030** | **0.1183** | **0.1765** | **0.0130** | **100.0** |
| URDFormer | 1.2288 | 1.2309 | 0.2914 | 0.4285 | 0.7128 | 0.8995 | 0.2840 | 4.48 |
| NAP-ICA-avg | 1.0280 | 1.0333 | 0.1771 | 0.2350 | 0.1144 | 0.1991 | 0.0222 | 18.59 |
| Ours-avg | **0.9528** | **0.9565** | **0.1530** | **0.1989** | **0.1040** | **0.1701** | **0.0087** | **39.34** |

**More qualitative results.** We show more examples of our method tested on real images from the internet in Figure 10. We observe that our method can robustly generate plausible objects from real images with variations in appearance in terms of part arrangement and object texture, and even in the presence of backgrounds (as shown in the last two rows). We provide more qualitative results showing generated objects in animation for our model and other baselines *in the supplementary video*. The video shows 6 examples each for the PartNet-Mobility, ACD dataset, and the real images. We observe that our model can generalize well to the unseen and real images of objects with variational appearances in terms of part arrangement and object texture.

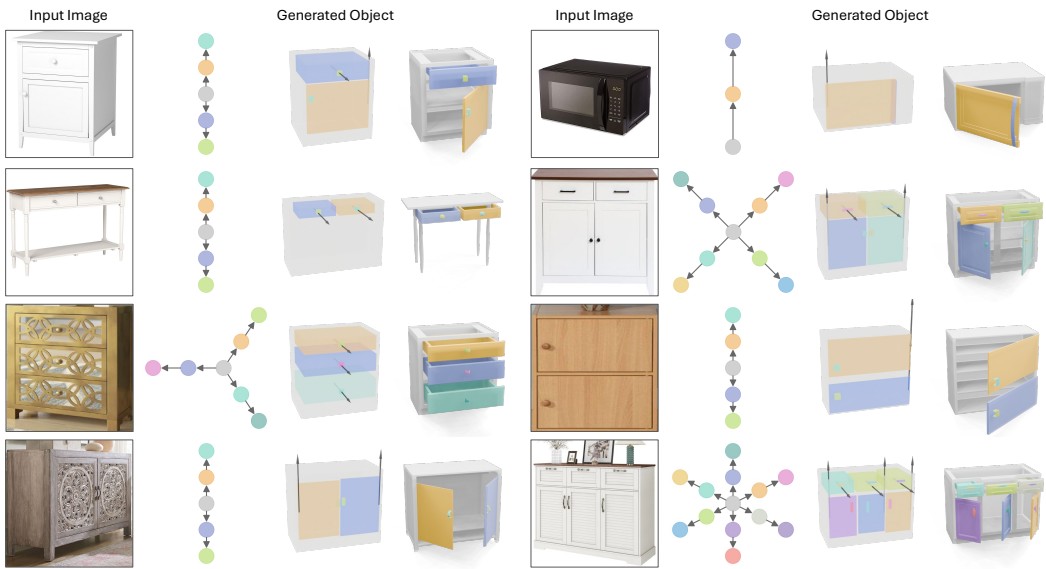

Figure 10: Examples of our method tested on real images from the internet. Our method can robustly generate plausible objects from real images with varied appearance in terms of the part arrangement and object texture, and also when there is a background in the image (see last two rows).

**Editability of the generated objects.** We design a staged approach to enable human intervention and to incorporate user changes for interactive generation. Figure 11 demonstrates the editability of the generated results by showing the process of modifying the graph and specifying the part attributes in the graph. We observe that our model can generate plausible objects that are consistent with the user's intention. Even when the user-specified graph asks the model to generate fewer or more parts than the image suggests, our model can leverage the learned prior of articulated objects to make the necessary adjustments to the object to respect the user's intention while maintaining object plausibility.

**Test on images with objects in arbitrary state.** Our model is trained on images of objects in a resting state. To evaluate the generalization of our model on unseen states, we test our model on images of objects in arbitrary states in a zero-shot setting. Specifically, we feed the model with renderings of the testing objects in the PartNet-Mobility dataset by randomizing articulated states and viewpoints. We show the qualitative results in Figure 12. We observe that our model is still able to generate plausible objects when the parts are with reasonable amount of articulation and visibility, as shown in the cases in the middle column. However, our method struggles to produce consistent results with the input image when the parts are in extreme states (e.g., with doors widely opened), as shown in the cases in the right column. The graph prediction from GPT-4o can be inaccurate in these cases due to the existence of self-occlusion. Our generative module presents difficulties in understanding the spatial structure of the parts under large differences in object shape from the training data. The revealed interior of the objects (e.g., the shelves inside the cabinet) can also be a distraction for the model. We also observe that our model cannot produce the objects with the same articulation as the image shows, which can be an interesting direction for future work.

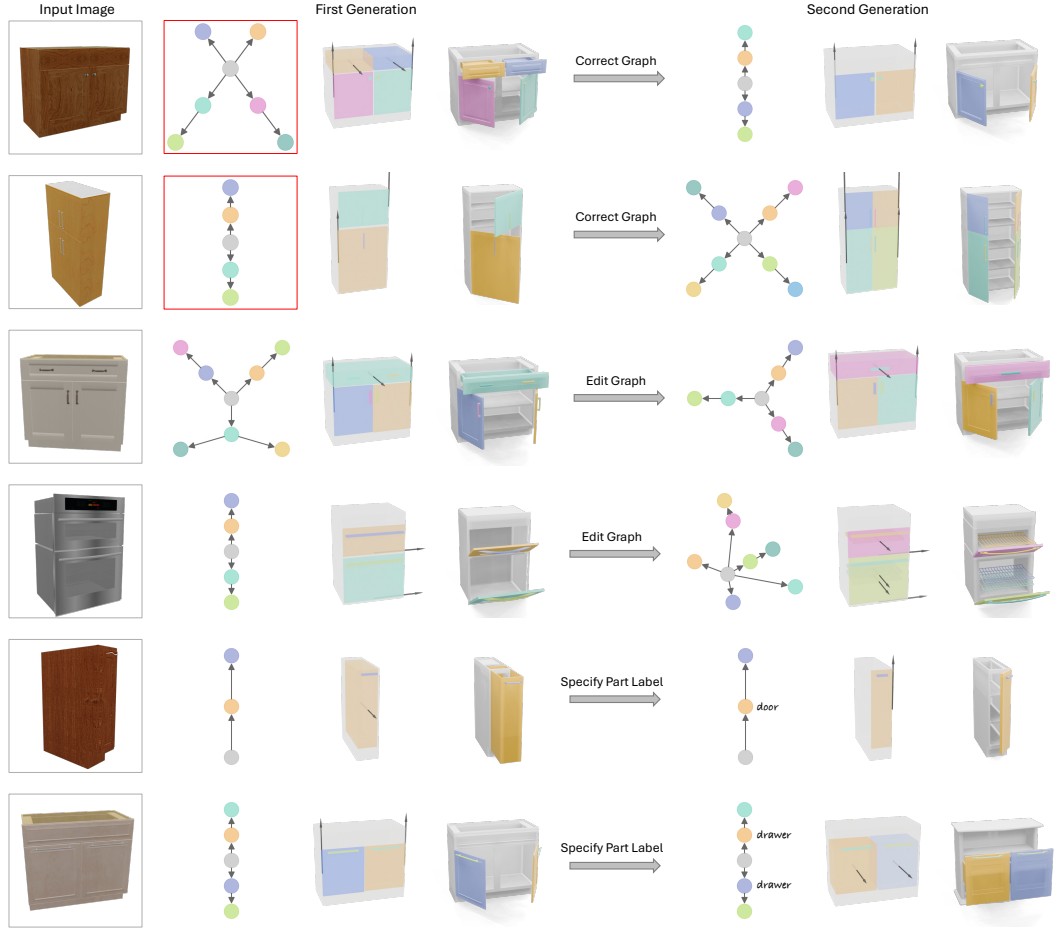

Figure 11: Examples demonstrating how our method enables interactive generation by user-driven graph correction (see rows 1 and 2) or graph editing (see rows 3 and 4) and specifying the part attributes in the graph (see rows 5 and 6). Our model generates plausible objects that are consistent with the user's intention. Even when the user-specified graph asks the model to generate fewer or more parts than the image suggests (as shown in rows 3 and 4), our model can make the necessary adjustments to the object to respect the user's intention while maintaining plausibility.

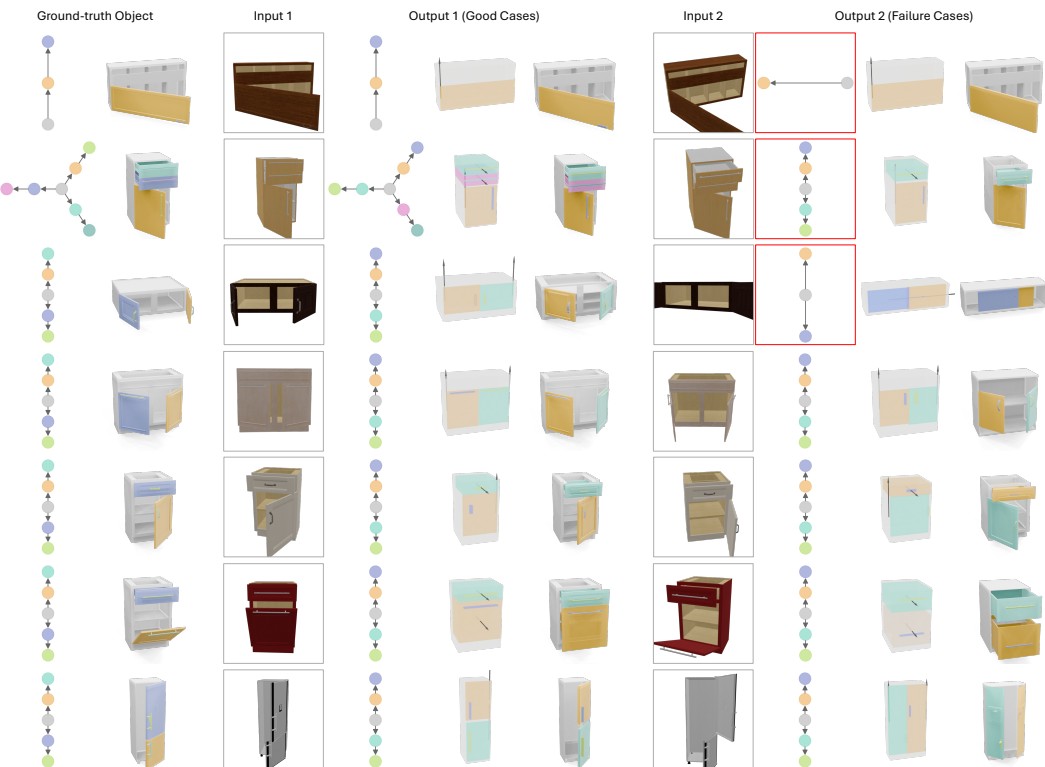

Figure 12: Examples of our method tested on images of objects with arbitrary articulated states from random views. We observe that our model is still able to generate plausible objects when the parts are with reasonable amount of articulation and visibility, as shown in the cases in the middle column. However, our method struggles to produce consistent results with the input image when the parts are in extreme states (e.g., with doors widely opened), as shown in the cases in the right column.

