# OpenReview forum: "SINGAPO: Single Image Controlled Generation of Articulated Parts in Objects"
_ICLR.cc/2025/Conference — ICLR 2025 Poster_

### Official Review · Reviewer_Do48 · 2024-11-03

**Soundness:** 3
**Presentation:** 3
**Contribution:** 3
**Rating:** 6
**Confidence:** 3

**Summary:**

This paper presents a method for generating 3D articulated objects from a single image, addressing the limitations of prior work that require multi-view or multi-state inputs. Observing an object in a resting state, the proposed approach leverages a diffusion model to capture plausible variations in geometry and kinematics, effectively handling the ambiguity posed by single-view input. The generation pipeline progresses from coarse structure to fine geometric detail, using a part connectivity graph and part abstraction to guide the process. Experiments show that this method outperforms existing approaches in realism, input-image consistency, and reconstruction quality, marking a significant step towards scalable and practical 3D articulated object creation for applications in virtual environments and robotics.

**Strengths:**

1.	The problem tackled in this paper holds substantial practical significance. In contrast to prior approaches that necessitate multi-view or multi-state inputs, this work enables the reconstruction of articulated 3D assets from a single image, thereby significantly enhancing the generalizability and applicability of the method.
2.	This paper effectively deploys various features to guide the generation process, employing a staged architecture that progresses from coarse to fine detail. The experimental results validate the effectiveness of this staged generation approach in producing high-quality articulated objects.
3.	This paper tackles the ambiguity of generating 3D articulated objects from single-view images, addressing challenges like occlusion and limited visual cues for movable parts. A part connectivity graph guides the spatial layout and articulation hierarchy, ensuring coherence even for small, obscured parts. This approach effectively balances realism and structural plausibility, advancing scalable 3D asset creation from minimal input.

**Weaknesses:**

1. The proposed method is mainly applied to specific categories of objects (such as cubic furniture and household appliances), and may not be directly applicable to objects with more complex shapes and more diverse categories (such as irregular shapes or soft structures). This limits the versatility of the method.

2 The use of GPT-4o for part connectivity graph generation faces challenges such as ambiguity handling, prompt sensitivity, and generalization, which are not addressed in the paper. With single-view input, GPT-4o may struggle with occlusion or complex structures, leading to inaccuracies in graph generation. The method's reliance on specific prompts can vary in effectiveness depending on object types and viewpoints, yet the authors do not explore mitigation strategies. Additionally, GPT-4o's performance is limited by its pretraining data, affecting generalization to unseen categories. Addressing these limitations would enhance the method's robustness and applicability in scenarios with high structural variability.

**Questions:**

1.The article mentions that the coarse-to-fine generation method gives users more editability, but the article does not explain or experiment on this point. Does this mean that users need to be given more interactive experience in the generation stage, similar to text-guided generation?

---

> ### Author Response · Authors · 2024-11-21
>
> Thank you for appreciating the importance of our work and providing insightful feedback.
>
> **W1** & **W2** & **Q1**: Please refer to the general response.

---

> > ### Comment · Reviewer_Do48 · 2024-12-01
> >
> > Thanks for the authors replay, It addresses my concerns, and I will maintain my scores with a leaning towards acceptance.

---

> > > ### Author Response · Authors · 2024-12-02
> > >
> > > Thank you for your time and positive feedback. We're glad that our response addressed your concerns.

---

### Official Review · Reviewer_eFff · 2024-11-03

**Soundness:** 3
**Presentation:** 3
**Contribution:** 3
**Rating:** 6
**Confidence:** 3

**Summary:**

In this work, the authors proposed a new framework called SINGAPO to generate articulated objects from a single image. The framework designs a modular way to create objects from coarse to fine, including the connectivity graph, attributes to describe the articulated parts at the abstract level, and 3D mesh. The method uses a diffusion model in the process of generating abstract attributes, which allows for variations to account for any ambiguity in the image. The authors also point out that unlike previous methods that require multi-view or multi-state inputs and methods that only allow coarse control over the generation process, this method is the first to explore the generation of articulated objects from a single image.

**Strengths:**

- Overall, the paper is well-organized and easy to understand. The motivation is clear.
- The paper considers the situation that multiple reasonable articulations of objects may exist when there is only one image input, which is very important for this problem.
- The experimental results show that the proposed method achieves encouraging results on multiple datasets, and the visualization results are reasonable.

**Weaknesses:**

- Among the limitations mentioned in the paper, it seems that the method has difficulty handling cases with complex textures and cases with complex arrangements. Does this also mean that the method has difficulty handling inputs of real objects? The paper also does not show results with real pictures as input. This raises some concerns about the generalization ability of the model.
- Since the meshes generated by the model come from retrieval, it will not work if there are some shapes in the input image that are not in the model library.
- This method relies too much on the correct connection graph from the input and can not correct the wrong connection graph. This somewhat weakens its claim of "generating articulated objects from a single image".

**Questions:**

- Quantitative results show that $L_{fg}$ is effective. Could you also visualize the attention map to show what kind of attention map the model learns if the $L{fg}$ is removed?
- Could you provide some results of real image inputs?
- The paper points out the lack of control over the generation process in previous methods. But I don't understand how / if the method can control the model to switch between multiple states due to the ambiguity in the image (shown in Fig. 1). These states are reasonable and the connectivity graph is the same, how can I control it or change the generation results?
- And are these ambiguities taken into account during evaluation? (For example, generate multiple possible states during quantization?)

---

> ### Author Response · Authors · 2024-11-21
>
> Thank you for appreciating the importance of our work and providing insightful feedback.
>
> **W1 "Results on real pictures?"**: Please see the general response and refer to Fig. 10 for qualitative results.
>
> **W2 "The retrieval won't work if some shape in the image are not in the model library"**: Our primary focus is not on detailed geometry reconstruction but on addressing the challenge of recovering the geometric and kinematic structure of articulated objects from a single input image. We agree that improvements in detailed geometric reconstruction are a promising direction to investigate, but this is beyond the scope of our paper.
>
> **W3 "Cannot correct the wrong graph"**: We design the graph extraction as an intermediate step so that the user can easily correct the graph and control/edit the output object in an interactive generation fashion. In contrast, prior work addresses the task by producing the graph and other shape/motion attributes altogether. Their end-to-end design limits user intervention, making it difficult to re-generate the output by incorporating changes. Please also refer to the general response for further discussion.
>
> **Q1 "What will the attention map look like when removing $L_\text{fg}$"**: The visualization of the attention map is shown in Fig. 11 in the appendix. We observed that without this loss, the attention tends to be scattered and occasionally distracted by the noisy feature in the background.
>
> **Q2**: See response for W1.
>
> **Q3 "How to switch between different outputs due to the ambiguity? How to change the generation results? "**: The ambiguity in geometry and articulation (as shown in Fig. 1) is reflected in a distribution over the output objects. To get multiple outputs with variations, different random noise initializations can be used to produce multiple outputs, from which the user can select the desired ones. Or the user can specify the attribute (e.g., semantic label) for certain parts to control the generation, as shown in Fig. 12.
>
> **Q4 "Are the ambiguities taken into account during evaluation?"**: For quantitative evaluation, each test image is associated with only one ground-truth object in the dataset, so our metric cannot evaluate whether a different output constitutes another plausible solution or not. To compensate for this challenge in quantitative evaluation, we conducted a user study to evaluate the overall realism based on human intuition (see line 458 in paper). We agree that it is valuable to take this ambiguity into account for future metrics.

---

> > ### Comment · Reviewer_eFff · 2024-11-25
> >
> > Thanks to the author for the reply, this solved my problems and I will keep my rating.

---

> > > ### Author Response · Authors · 2024-11-27
> > >
> > > Thank you for your time and positive feedback. We're glad that our response addressed your concerns.

---

### Official Review · Reviewer_WBnp · 2024-11-04

**Soundness:** 3
**Presentation:** 3
**Contribution:** 3
**Rating:** 8
**Confidence:** 4

**Summary:**

This paper proposes a framework for single image-conditioned 3D articulated object generation. The framework comprises 3 steps: (i) use a large vision-language models to predict the part connectivity graph from the image; (ii) generate attributes to describe the articulated parts at the abstract level; (iii) retrieve part meshes to assemble the final object.

**Strengths:**

- The paper addresses an important problem of image-conditioned articulated object generation, with detailed explanations of its importance and challenges.
- The experiments are thorough and well-designed. In addition to the main comparisons, the authors also showed a comparison with a multi-view method Real2Code and ablation studies of each module component as well as data augmentation. There is also a clear discussion of failure cases and limitations.
- Both qualitative and quantitative results look good. Compared to the baselines, the part graphs generated by the proposed method seem to have better global coherence.

**Weaknesses:**

The overall framework and experiments look good to me. Some minor points:
- If GPT-4o gives a wrong answer to the part graph, there will be no correction to this in the following stages? Intuitively, given an object image, how its part graph can be like should also depend on the prior distribution of part graphs of the object collection.
- As discussed in the introduction, one challenge for the task is that "the parts are small and thin, making them hard to perceive from a single image". The DINOv2 features are on 14x14 image patches, would that be too coarse for the small parts (like handles and knobs)?

**Questions:**

- For the dataset, when rendering the images, what part states are used? (e.g. door open or closed) It would be good to also explain this. As mentioned in the Introduction, "the object is observed in the closed state where occlusions introduce ambiguity in the part shape and articulation". In fact, it would be interesting to see for the same 3D articulated object rendered under different part states (different joint angles), how would the generation results be like (just an interesting bonus, without this I think the experiments are already quite comprehensive).

---

> ### Author Response · Authors · 2024-11-21
>
> Thank you for appreciating the importance of our work and providing insightful feedback.
>
> **W1 "No graph correction? The graphs depend on the prior distribution of the object collection?"**: Please refer to the last point in the general response for the graph correction mechanism. It’s unknown how the prior knowledge is used by GPT-4o for graph prediction, but we prompt it to recognize all the parts from the image first, and then connect them (see Appendix A.2). We observe that errors are mainly due to over- or under-recognition of parts. “Invalid part connection” errors such as connecting two drawers together are uncommon.
>
> **W2 "Would the 14x14 image patches be too coarse for small parts?"**: We observed empirically that 14x14 patches work well when parts are not too cluttered to be distinguished. We attribute this to the use of DINOv2 for the detailed feature extraction. We agree that the patch resolution could be a bottleneck when generalizing to more complex objects (e.g., with a much higher number of parts and highly compact part arrangement), especially when small and thin parts considered.
>
> **Q1 "Clarify the rendering state in the dataset. What if the object is under different states?"**: Thank you for the suggestion. We have clarified in our revision (in line 339) that we render the resting state of the objects from the dataset to train our model. As suggested, we tested our model on images of objects in arbitrary articulated states in a zero-shot manner. Please see Fig. 13 in the appendix. Overall, we generate plausible objects when the parts exhibit a reasonable amount of articulation and part visibility. However, our approach struggles to produce consistent results with the input image when the parts are in extreme states (e.g., with doors widely opened).

---

> > ### Comment · Reviewer_WBnp · 2024-12-02
> >
> > Thanks for the explanations!
> >
> > I want to say that I really like the experiments in this paper. I usually read papers in a way that I first read all the sections before the experiments and think about what experiments I would expect to see to justify all the points claimed in these sections. When reading this paper, all experiments and ablations I thought about can be found in the paper, including some that I may just suggest as additional (that means if a paper doesn't have such an experiment I'll only write as a minor suggestion instead of a weakness) -- especially the ablations and discussions of failure cases, which I think is very important but many papers don't put much effort on nowadays. I really appreciate these solid experiments done by the authors.

---

> > > ### Author Response · Authors · 2024-12-02
> > >
> > > Thank you so much for your thoughtful and positive feedback. We're glad that our experiments and discussions aligned with your expectations. We appreciate your recognition of our efforts, and your kind words are truly motivating.

---

### Official Review · Reviewer_7AKv · 2024-11-04

**Soundness:** 3
**Presentation:** 3
**Contribution:** 2
**Rating:** 6
**Confidence:** 4

**Summary:**

This paper proposes a novel method for generating articulated objects from a single image. Given an image of an object in its rest state, it first asks ChatGPT-4o to output the object's connectivity graph. It then trains a diffusion model to generate coarse descriptions of the parts, conditioned on the input image and the connectivity graph. Finally, it retrieves similar parts from a large database to generate the concrete geometry of each part.

**Strengths:**

1. The idea of training a diffusion model to generate parts conditioned on a part graph and input image is interesting, as it enables the generation of variations.

2. The authors demonstrate that the proposed method outperforms the baseline and provide extensive ablation studies.

3. Overall, the paper is presented clearly.

**Weaknesses:**

1. The overall coarse-to-fine pipeline for articulated object generation is not uncommon.

2. The proposed method only handles closed-domain objects and relies on ground-truth annotations for training. However, 3D datasets with fine-grained part annotations are extremely challenging to annotate. In the experiments, the authors focus on only six part categories and seven object categories, which may significantly limit the practical applicability of the proposed method in much more diverse and complex real-world scenarios.

3. The inference of the part connectivity graph relies on GPT-4o, which lacks specific designs and may not be considered a major contribution of the paper. It may also be difficult to inject 3D bias or other priors into the pre-trained GPT-4o.

4. The detailed geometry of parts is retrieved from an existing database. There are concerns about the visual quality and consistency of the parts, which may not be effectively captured by current quantitative metrics like IoU and Chamfer distance. Perceptual metrics may provide better ways to evaluate visual quality and consistency. Can the module retrieve complex and fine-grained parts in addition to primitive-like parts?

Additionally, this retrieval component relies heavily on and is adapted from a previous work.

5. According to Table 2, some components bring minimal gains, such as the classifier-free training on the object category (cCF) and the foreground loss \(L_{fg}\).

**Questions:**

1. What does `r_i` represent in line 200?

2. I am unclear about the phrase, "the graph determines the ordering of the parts to be generated so that the parts are organized correctly when articulating." Could you explain this in more detail?

3. In line 331, could you clarify why "graph accuracy" for "NAP-ICA-GTgraph" is not 100%?

4. Line 421: The sentence "we report the accuracy (Acc% ↑) in terms of the percentage of correct graph topology" is unclear to me. Could you explain this further?

5. Do you use one token for each part in your transformer?

6. What is the difference between "image guidance (iCF)" and "image cross-attention (ICA)"?

---

> ### Author Response · Authors · 2024-11-21
>
> Thank you for recognizing the value of our paper and providing insightful feedback.
>
> **W1 "Coarse-to-fine is not uncommon"**: Prior work produces the graph and other shape/motion attributes at once. This design limits user controllability/editability, and makes it difficult to correct errors or incorporate changes and re-generate the output. The general concept of a coarse-to-fine strategy is not new, but to the best of our knowledge, we are the first to introduce such a pipeline for image-conditioned articulated object generation.
>
> **W2 & W3**: Please refer to the general post above.
>
> **W4 "Can the module retrieve complex and fine-grained parts?"**: Our focus is not on detailed geometry reconstruction but on recovering the geometric and kinematic structure of articulated objects from a single image. As part geometries in household articulated objects are often structurally aligned and semantically shared, we adapted a retrieval approach based on structural and semantic matching of parts that is similar to prior work. This approach works fairly well with few cases where the output is unrealistic due to flaws in the part geometry. We agree with the reviewer that further improvements in part geometry reconstruction is a promising direction to investigate.
>
> **W5 "Minimal gains on `cCF` and $L_\text{fg}$"**: We add `cCF` so that our model avoids taking the object category as a mandatory condition. Having this category-agnostic training helps learn common features across object categories, which contributes to better generalization on unseen data, as shown in Table 4. We add the $L_\text{fg}$ loss to minimize the distraction of the noisy DINOv2 features in the background. We visualize the attention map in Fig. 11.  Its contribution to the overall loss in our ablation is less prominent because our testing images have pure background and the noise is not significant.
>
> **Q1 "What does `r_i` represent?"**: `r_i` is the motion range for each part. Thank you for pointing this out. We have fixed the typo in line 202.
>
> **Q2 "Explain the phrase in more detail"**: What the diffusion model outputs is essentially a list of part attributes, where each part corresponds to a node in the graph. The ordering of parts in the list is in accordance with the conditional graph represented as an adjacency matrix. The model generates parts in the order indicated by the graph, with relationships between the generated parts accordingly respected. For example, a root node in the graph should not contain the attributes of a handle.
>
> **Q3 "Why graph accuracy of `NAP-ICA-GTGraph` is not 100%?"**: The difficulty of controlling the output using a graph is a known weakness of NAP that has been discussed by Liu et al. [1]. This happens because the graph topology produced by NAP is post-processed by a minimum spanning tree algorithm based on the size of generated parts, which can ignore the node-edge existence specified by the user input. In our experiments, we observe that specifying the graph for NAP as a condition can even worsen its performance.
>
> **Q4 "Unclear about the `Acc%` metric"**: We evaluate the correctness of the graph by checking whether its topology and edge directions exactly match that of the ground-truth graph. Then we report the percentage of the correct graph across the test data.
>
> **Q5 "Do you use one token for each part?"**: Each part attribute is a separate token, following our base model. The tokens have positional encodings so that their association with the part is captured.
>
> **Q6 "What is the difference between `iCF` and `ICA`?"**: `ICA` is the cross-attention module that injects the image condition into the diffusion model. We apply the classifier-free training for images `iCF` by dropping the image condition (by replacing the input patch feature with a dummy one), at a particular ratio. In this way, we can enforce stronger image guidance during inference by giving it higher weights as mentioned in line 303.
>
> [1] Liu, J., Tam, H. I. I., Mahdavi-Amiri, A., & Savva, M. (2024). CAGE: Controllable Articulation GEneration. In *Proceedings of the IEEE/CVF Conference on Computer Vision and Pattern Recognition* (pp. 17880-17889).

---

> > ### Comment · Reviewer_7AKv · 2024-11-27
> > **Thank you**
> >
> > I appreciate the authors' response, which has addressed some of my questions. While I still have concerns about the novelty, the significance of the contributions, and part retrieval (e.g., consistency), I am not opposed to accepting the paper. I have raised my rating to borderline accept and hope the authors can clarify these points in their revision.

---

> > > ### Author Response · Authors · 2024-11-27
> > >
> > > Thank you for your time and positive feedback. We're glad that our response addressed your concerns. We will further clarify those points in our camera-ready revision.

---

### Author Response · Authors · 2024-11-21
**General Response for Rebuttal**

We thank the reviewers for the thoughtful and constructive feedback.

We are glad the reviewers found our work to address an important problem ($R_\text{WBnp}$) that holds substantial practical significance ($R_\text{Do48}$); our idea of training a diffusion model for this task to be interesting as it enables the generation of variations ($R_\text{7AKv}$) that consider multiple reasonable articulations of objects due to the ambiguity of a single image input ($R_\text{eFff}$); our experiments to be thorough and well designed ($R_\text{WBnp}$) and to provide extensive ablation studies ($R_\text{WBnp}$, $R_\text{7AKv}$); and our results to validate the effectiveness of the approach in producing high-quality articulated objects ($R_\text{Do48}$) and outperform the baseline ($R_\text{7AKv}$).

We revised our paper to incorporate reviewer feedback (revisions are denoted in purple text). In this post we respond to common questions from reviewers.  Individual reviewer questions are answered under each review. We are happy to respond to any additional questions and welcome further suggestions or updated opinions from the reviewers.

**How does our pipeline enable controllability ($R_\text{eFff}$) and editability ($R_\text{Do48}$)?**

Please see Fig. 12 in the appendix for examples showing how users can correct or edit the results for interactive generation.

**Examples of taking real images as input ($R_\text{eFff}$)?**

We showed some results on real images at the end of the supplementary video. In this revision, we add more results in Fig. 10 of the appendix. Overall, we did not observe a large sim-to-real gap when testing on real images. Our diffusion model consumes high-dimensional DINOv2 features instead of the original image, and performs similarly on real and synthetic images showing objects of comparable complexity.

**Generalization to diverse and complex objects beyond specific categories ($R_\text{7AKv}$, $R_\text{Do48}$)?**

We propose an approach for image-conditioned generation of articulated objects modeled to the level of actionable parts, which itself is a challenging task involving highly structured data. We select object categories with associated part categories to demonstrate the effectiveness of our method, but our approach is agnostic to object categories. The object categories we selected cover a high portion of articulated everyday objects, and have been used consistently in prior work. Furthermore, we did not make any strong category-specific assumptions that prevent extending our approach to other object categories.  Most importantly, the part categories in our approach cover the bulk of articulations observed in everyday objects.

**Does using GPT-4o limit generalization ($R_\text{7AKv}$, $R_\text{Do48}$), and is there a correction mechanism for graph errors ($R_\text{WBnp}$**, **$R_\text{eFff}$)?**

Customizing a large multi-modal model is effective in our task, especially given that articulated object data are sparse. As shown in Tables 1 and 2, GPT-4o demonstrates good performance in recognizing parts and reasoning about part connectivity when provided with a few examples and appropriate instructions (see Appendix A.2). As for corrections of errors in graph inference, we design a staged approach precisely so that the user can correct the graph in an interactive generation fashion. Please refer to Fig. 12 for examples.

---

### Meta-Review · Area_Chair_yJDR · 2024-12-07

**Metareview:**

In this paper, the authors have proposed SINGAPO, which aims to generate articulated objects from a single image. The motivation is clear, the problem it gargets at is important, the paper is well-written, and the experiments are comprehensive. The reviewers initially have some concerns but they are well-addressed by the rebuttal. All the reviewers agree to accept the paper. I recommend a decision of acceptance.

**Additional Comments On Reviewer Discussion:**

The reviewers initially have concerns on additional experiments, the pipeline of the method, and the generalization ability of the model. The authors have addressed the concerns in the rebuttal and the reviewers are satisfied with the reply.

---

### Decision · Program_Chairs · 2025-01-22

Accept (Poster)